# Novel Metabolomic Approach for Identifying Pathology-Specific Biomarkers in Rare Diseases: A Case Study in Oculopharyngeal Muscular Dystrophy (OPMD)

**DOI:** 10.3390/metabo13060769

**Published:** 2023-06-19

**Authors:** Pradeep Harish, Alberto Malerba, Rosemarie H. M. J. M. Kroon, Milad Shademan, Baziel van Engelan, Vered Raz, Linda Popplewell, Stuart G. Snowden

**Affiliations:** 1Department of Pharmacology and Therapeutics, Institute of Systems, Molecular and Integrative Biology, University of Liverpool, Liverpool L69 3GE, UK; 2Department of Biological Sciences, Royal Holloway University of London, Egham TW20 0EX, Surrey, UK; 3Department of Rehabilitation, Donder Institute for Brain, Cognition and Behaviour, Radboud University Medical Centre, 6525 AJ Nijmegen, The Netherlands; 4Department of Human Genetics, Leiden University Medical Centre, 2333 ZC Leiden, The Netherlands; 5National Horizons Centre, Teesside University, Darlington DL1 1HG, County Durham, UK

**Keywords:** biomarker, metabolomics, LC-MS, random forest, oculopharyngeal muscular dystrophy

## Abstract

The identification of metabolomic biomarkers relies on the analysis of large cohorts of patients compared to healthy controls followed by the validation of markers in an independent sample set. Indeed, circulating biomarkers should be causally linked to pathology to ensure that changes in the marker precede changes in the disease. However, this approach becomes unfeasible in rare diseases due to the paucity of samples, necessitating the development of new methods for biomarker identification. The present study describes a novel approach that combines samples from both mouse models and human patients to identify biomarkers of OPMD. We initially identified a pathology-specific metabolic fingerprint in murine dystrophic muscle. This metabolic fingerprint was then translated into (paired) murine serum samples and then to human plasma samples. This study identified a panel of nine candidate biomarkers that could predict muscle pathology with a sensitivity of 74.3% and specificity of 100% in a random forest model. These findings demonstrate that the proposed approach can identify biomarkers with good predictive performance and a higher degree of confidence in their relevance to pathology than markers identified in a small cohort of human samples alone. Therefore, this approach has a high potential utility for identifying circulating biomarkers in rare diseases.

## 1. Introduction

A biomarker can be thought of as any measurement reflecting an interaction between a biological system and a pharmacodynamic response to a chemical, physical or biological challenge. The measured response may be physiological, encompassing cellular biochemistry or a molecular interaction, or functional [1,2]. Biomarkers are often used to assess underlying biological processes and are vital tools in clinical practice as they allow clinical and pre-clinical diagnosis and the assessment of disease severity, progression and response to treatment [3,4,5].

The metabolome represents the ultimate consequence of the interaction between genes and the environment, making it sensitive to change and thus an ideal biochemical pool for detecting subtle changes associated with disease progression [6,7]. The ideal biomarkers associated with disease progression will be those that are causally linked to pathology rather than those that change as a result of it. However, the sensitivity of the metabolome to change also makes it susceptible to confounding factors. The blood metabolome is especially susceptible to these confounding factors as the composition is influenced by multiple systemic factors, including diet and the functioning of all body organs [8,9]. Traditionally, metabolomics studies looking to robustly study peripheral metabolism have tackled this issue by analyzing large cohorts of patients and performing a comparison to healthy controls with potential biomarkers being subsequently validated in an independent sample cohort [10,11]. However, in rare diseases, the paucity of sample availability, the geographic distribution of patient groups, and varying methods of sample collection renders the above approach logistically, financially and scientifically unviable. Hence, there is an urgent need to develop novel approaches for identifying metabolic biomarkers in rare diseases that mitigate these unique challenges posed by such pathologies.

The composition of the peripheral metabolome is heavily affected by confounding influences [8,9]; the closer the sample is to the site of pathology, the stronger the disease-related metabolic perturbations will be and the weaker the systemic factors, making it easier to identify the metabolic signature of a disease. In this project, we used a combination of tissue and peripheral samples to develop a method for identifying metabolic biomarkers in rare diseases using oculopharyngeal muscular dystrophy (OPMD) as a model. OPMD is an autosomal dominant progressive degenerative muscle disorder caused by mutations in the poly(A)-binding protein nuclear-1 (PABPN1), which is crucial for the proper processing of mRNA. Despite recent advances in molecular techniques and the elucidation of disease mechanisms [12,13], the low rate of prevalence coupled with variable age of onset of the disease, a lack of familiarity with disease presentation and a lack of capacity for genetic testing have contributed to poor diagnosis in population clusters not thought to be at risk. While the gold standard for identifying inherited rare diseases is PCR-based assays, this approach would benefit other analyses of new biomarkers to define the onset of a disease. Additionally, there is an urgent need for quantitative, high-throughput and longitudinal biomarkers that can diagnose disease and monitor both pathological progression and therapeutic response over the course of a clinical trial as well as the patient lifetime [14]. Such a biomarker, when translated into an assay that utilizes already established techniques (e.g., colorimetric ELISAs), will enable the local pathology lab to run samples without the need for expensive equipment or external partners. This will rapidly speed up disease diagnosis and achieve more accurate monitoring. 

While the ideal biomarker characterization experiment will involve comparing the pathology-specific metabolic profiles from affected human tissues and peripheral blood (and the equivalent for healthy human controls), attaining such tissue samples poses ethical and logistical challenges to the laboratory-based scientist. Hence, there is a need to integrate biomarker pipelines across species with pre-clinical models (which can be generated in enough numbers for high-throughput experiments) bearing the statistical burden associated with a discovery experiment and thereby reducing the need for an extensive set of patient samples. 

In keeping with our pipeline, we will initially identify a pathology-specific metabolic fingerprint in murine (model) tissue. It is our hypothesis that as the muscle is the site of pathology in OPMD, primary disease-specific effects will be enriched in the mouse muscle metabolome. As the blood receives secreted metabolomic products from all major organ compartments, our next objective will be to model how these disease-specific shifts from muscles become “diluted” in the mouse blood. This will enable the identification of a panel of candidate biomarkers (CBMs) that can subsequently be validated in human plasma samples collected from OPMD patients and healthy controls. To ensure translational applicability, clinical relevance and a human context, the final step involves validation in a smaller set of patient populations (as compared to a full discovery cohort).

## 2. Materials and Methods

### 2.1. Generation of Murine Samples

The A17.1 model mouse for OPMD has been characterized previously and used routinely by groups to develop therapies for the disease [15,16,17,18]. Briefly, the model encompasses the overexpression of a bovine *expPABPN1* gene specifically in skeletal muscles. The model appears to manifest overt pathological changes at around 12 weeks of age. To investigate metabolic changes associated with the disease before the onset of pathological burden and after, we analyzed samples at two different time points: 4 weeks of age and 24 weeks of age. 

Animal experiments were performed at Royal Holloway, University of London, under the auspices of ASPA (1986), UK Home Office project license P36A9994E, and approved by the university’s animal welfare and ethics review board. A17.1 OPMD disease model mice and FvB healthy littermate controls were housed with food and water ad libitum in minimal disease facilities. Mice were generated by breeding heterozygous males with WT females. Due to the heterozygous nature of the disease model, OPMD mice were analyzed to confirm the genotype by PCR, with primers directed against the bovine *PABPN1* insert (5′-GAACCAACAGACCAGGCATC-3′ and 5′-GTGATGGTGATGATGACCGG-3′). The PCR cycle implemented an initial denaturation at 95 °C for 2 min, followed by 40 cycles of 95 °C denaturation, 60 °C for annealing, and 72 °C for extension, with each step lasting for 30 s. The final extension was conducted at 72 °C for 10 min and products visualized using agarose gel electrophoresis. 

Four-week-old A17.1 OPMD model (9 animals) mice and littermate FvB healthy controls (8 animals) were fasted overnight, sacrificed, with blood collected by cardiac puncture and allowed to clot overnight at 4 °C. The serum was extracted and spun down successively with increasing speeds (1000× *g*, 2000× *g* and 3500× *g*) to remove residual cells. The serum sample thus collected was stored at −80 °C. Gastrocnemius samples collected from mice were flash-frozen in liquid nitrogen and then stored at −80 °C until further analysis. Additional untreated 24-week-old gastrocnemius samples from a previous study were also utilized [16] as an additional time point.

### 2.2. Oculopharyngeal Muscular Dystrophy Patient Samples

Patients were recruited through the Dutch neuromuscular database (Computer Registry of All Myopathies and Polyneuropathies: CRAMP) [19,20]. All participants signed informed consent and the study was approved by the local ethics committee. All patients visited the outpatient clinic at the Radboud University Medical Centre and were clinically examined on swallowing function, the presence of ptosis and muscle weakness. A summary of clinical details is found in Table 1; blood was collected from anonymous controls.

### 2.3. Chemicals and Reagents

All solvents and reagents used for LC–MS analysis (water, methanol, acetonitrile, methyl-tertiary butyl ether (MTBE) and ammonium formate) were all LC–MS grade and purchased from either Sigma-Aldrich (St. Louis, MS, USA) or Fisher Scientific (Hampton, NH, USA). Two internal standards were added to all samples: L-valine ^13^C_5_^15^N (95%) and tripentadecanoin; both were purchased from Sigma-Aldrich.

### 2.4. Preparation of Tissue and Plasma Samples

Muscle tissue was homogenized using with a 4 mm steel ball bearing and 10 µL of methanol:water (80:20 *v*:*v*) per milligram of tissue added to each muscle sample prior to mechanical homogenization at 25 Hz for 10 min. Then, 25 µL of homogenate or 15 µL of plasma was transferred to an Agilent HPLC Vial with a 250 µL glass insert. Samples were then extracted using a modified version of an in-vial dual extraction described previously [21,22]. Briefly, 5 µL of HILIC internal standard (2.5 mM L-valine 13C515N in 80:20 MeOH:H2O), 140 µL of MTBE containing 15 µM of tripentadecanoin and 40 µL of methanol were added to all cell pellets and left to stand for 10 min to disrupt the cell membranes. After this, samples were transferred to an HPLC vial with a 250 µL glass insert, and 30 µL of water containing 0.15 mM ammonium formate was added before the sample was spun at 5000× *g* for 5 min to achieve phase separation. Extraction blanks were obtained by extracting 15 µL of HPLC-grade water using the same protocol, and 5 µL of all analytical samples was pooled to create quality controls. 

### 2.5. LC–MS Analysis of Aqueous Phase (HILIC)

LC–MS analysis was performed on an Agilent Infinity HPLC system coupled to an Agilent 6550 ion funnel QToF (Agilent, Santa Clara, CA, USA). Separation of the aqueous phase metabolites was performed on Agilent Poroshell HILIC-z column (2.1 × 100 mm, 2.7 μM) using 10 mM ammonium formate in water as mobile phase A and 2.5 mM ammonium formate in acetonitrile as mobile phase B. The column temperature was set to 30 °C with a flow rate of 0.25 mL/min, the mobile phase was isocratic for the first minute at 5% mobile phase A prior to a linear gradient increase to 10% at 6 min and 25% at 15 min after which there was a 3 min column washing step at 80% mobile phase A before initial conditions were restored to allow 7 min for column re-equilibration. Data were collected between 50–1000 m/z, with a gas temperature of 200 °C, a drying gas flow of 15 L/min, a nebulizer pressure of 40 psi, a sheath gas flow of 12 L/min and a temperature of 300 °C.

### 2.6. LC–MS Analysis of Non-Aqueous Phase (Reversed Phase)

Separation of the aqueous phase metabolites was performed on Agilent Poroshell C18 column (2.1 × 150 mm, 4.0 μM) using 10 mM ammonium formate in water as mobile phase A and 10 mM ammonium formate in methanol:MTBE (2:1 *v*:*v*) as mobile phase B. The column temperature was set to 55 °C with a flow rate of 0.425 mL/min, the gradient started at 20% mobile phase A before a linear decrease to 7% at 13 min, 6% at 20 min and 4% at 24 min prior to a 6 min washing step of 100% mobile phase B prior to the restoration of initial conditions to allow 5 min of re-equilibration. Data were collected between 50–1200 m/z, with a gas temperature of 200 °C, a drying gas flow of 15 L/min, a nebulizer pressure of 35 psi, a sheath gas flow of 10 L/min and a temperature of 120 °C.

### 2.7. Data Processing and Statistical Analysis 

All .d files generated by the MS were converted into .mzXML files using Proteowizard [23]. Converted data files were processed using the CAMERA package performed in the open source software package R (v3.6.0), with peak picking performed using a “centwave” method, which allows for the deconvolution of closely eluting or slightly overlapping peaks [24]. After this was performed, metabolite features were defined as any peak with an average intensity at least 5 times higher in analytical samples relative to the abundance seen in the extraction blanks, with the peak having to be present in all of the samples of at least one sample group.

The combined datasets (both HILIC and RP data) were analyzed using a range of multivariate algorithms including principal component analysis (PCA) and partial least squares discriminant analysis (PLS-DA) performed in SIMCA (v13.0.4) and with all data logarithmically transformed (base 10) and scaled to unit variance (UV). Model performance was assessed based on the cumulative correlation coefficients (R2X[cum]) and predictive performance based on 7-fold cross validation (Q2[cum]), with the significance of the model assessed based on the ANOVA of the cross-validated residuals (CV-ANOVA). Univariate analysis to identify individual metabolite variables that differed between groups was performed using generalized linear models (GLM) in R (v3.6.0). To test the predictive performance of the identified markers in human plasma samples, random forest models were calculated. The models were trained using all control samples (14) and 14 OPMD samples and were subsequently tested on the same 14 controls (owing to the limited number of these samples) and the remaining 37 OPMD patients.

## 3. Results

The protocol in this study was performed in four steps (Figure 1). In step 1, we perform initial analysis in the muscle tissue of 4-week old mice comparing A17.1 with control animals to identify metabolites associated with OPMD pathology where pathology has not overtly developed. In step 2, we validated the metabolites identified in step 1 in an independent set of mouse muscle samples collected from 28-week-old animals. Such muscles displayed overt pathology [16]. In step 3, candidate biomarkers (CBMs) were identified by translating the metabolites identified in steps 1 and 2 into plasma samples collected from the 4-week-old animals used for the initial discovery. In order to minimize the impact of secondary metabolic shifts linked to the disease, we opted to utilize plasma samples exclusively from 4-week-old mice to identify CBMs. This decision was based on the fact that the myopathy has a negative impact on metabolic homeostasis. As the disease advances and muscle mass decreases, secondary and tertiary effects associated with a loss of muscle metabolic flexibility become more prominent. In the final step, the identified CBMs were translated into human plasma samples to assess if they could predict the presence of OPMD pathology.

In this study, 327 of the metabolite features were unique to the murine muscle samples, with 365 and 1556 features unique to mouse and human plasma, respectively. In total, 1549 features were common to all 3 sample types; 231 were shared between only mouse muscle and plasma and 1182 were shared between murine and human plasma (Figure 2).

In the A17.1 mice model muscular dystrophy, initial metabolic signatures of disease were identified by comparing the muscle metabolite profile of 4-week-old healthy FvB controls against age-matched A17.1 mice using PLS-DA, which showed significant compositional difference (Figure 3A) with 326 metabolite features significantly different (*p* < 0.05). To ensure the robustness of this signature, metabolic perturbations were replicated in an independent set of FvB and A17.1 mice aged 28 weeks with 145 features validating between sample sets.

This signature was then translated into paired mouse serum with 1780 features measured in both mouse muscle and plasma matrices (Figure 4) with 20 of the 326 metabolite features successfully translating (*p* < 0.1), out of a total of 84 metabolite features that were significantly (*p* < 0.05) dysregulated in this matrix (Figure 3B and Table 2). We chose this less stringent threshold owing to the greater variability in the serum metabolome, relatively low sample number, and use as validation of previous findings.

These 20 metabolite features were subsequently translated into human patient plasma samples, of which 9 were shown to be significantly (*p* < 0.05) associated with OPMD sharing the same direction of change as was observed in the mouse serum samples (FvB > A17.1 & Control > OPMD representing a negative fold change, or FvB < A17.1 & control < OPMD representing a positive fold change in the disease state) (Figure 3C and Table 2).

Subsequently, PLS-DA models were re-calculated using only the 9 identified candidate biomarkers and were able to show clear separation in both muscle and serum samples collected from FvB and A17.1 mice (Figure 4A,B). These candidate biomarkers showed clear difference between OPMD patients and healthy controls (with a difference of 42.1% of t[1]); however, some overlap was observed (Figure 4C). As a result of this, random forest was applied to determine classification power of the CBMs (Figure 5), and after training, a sensitivity of 74% and a specificity of 100% (out-of-bag error of 18.3%) was achieved in the testing set, comprising 14 controls and 37 OPMD cases. However, owing to the limited availability of matched control samples (14), the controls in the testing set are the same controls used in the training set.

## 4. Discussion

Biomarkers are essential in clinical medicine as they enable the diagnosis of disease, the prediction of pathological progression as well as assessments of patient response to treatment. Ideally, these markers will be measured in peripheral fluids owing to the ease of collection, and are applicable for longitudinal assessments and global proliferation of analytical technologies across pathology labs. Furthermore, the blood metabolome, with its sensitivity to change, makes it the ideal biochemical pool for biomarker identification. However, the sensitivity that makes it ideal for identifying subtle changes associated with the early stages of disease is a double-edged sword as it makes the blood metabolome susceptible to confounding systemic effects [8,9].

To mitigate such an effect, and to initially investigate the abundance of primary pathology metabolic shift in peripheral tissues, we looked to determine the overlap in the metabolic signature of disease observed in muscle and paired serum samples; a total of 326 and 84 metabolite features significantly associated with OPMD pathology, respectively; however, only 20 were common to the two matrices. This result shows that pathology has a very different effect on metabolism in muscle compared to blood, meaning that it is not straightforward to generalize results obtained in the periphery to what is occurring in the pathologically affected tissue. This is an important observation as many metabolomics studies have measured a metabolic signature in the periphery to identify potential biomarkers and have generalized the findings to suggest that the identified markers are directly linked to pathology or the underlying biological process without any evidence to directly support this assumption [25,26], when in reality they could as easily be the result of systemic factors and hence secondary (or higher) effects of the disease state.

As stated previously, the best biomarkers are those that are causally linked to disease pathology as they enable specific diagnosis of disease as well as being able to accurately predict disease progression. This, combined with the limited overlap between the metabolic signature identified in the blood and muscle in this study, make it vital that future studies looking to identify biomarkers ensure that the identified candidates are directly linked to pathology [20]. This can be performed in two ways. The first is to use computational approaches such as Mendelian randomization [27,28]; however, this requires large sample numbers, making it unviable in rare diseases such as OPMD. The second way is to directly measure the metabolic signature at the site of pathology. However, in rare diseases, obtaining cohorts of tissue biopsy samples (e.g., muscle in OPMD, or retinae in ciliopathies) can be difficult. Hence, the pipeline proposed shifts the statistical burden of discovery to pre-clinical models, with the diagnostic power of these identified CBMs being subsequently tested in patients [20].

Using this pipeline, we have identified a panel of nine candidate biomarkers that correctly predicted the diagnosis of 81.6% OPMD patients. Whilst this is a relatively modest accuracy, an assay based on these biomarkers in the form of an “OPMD panel” could be used to rapidly identify high-risk individuals to be prioritized for genetic testing or to shed light on potential pathological trajectories to help inform clinical decisions on disease management. Even though this panel of CBMs has only been measured in a single, relatively small cohort, we can be more confident that they are both robust and specific biomarkers of OPMD than had we measured them in the small cohort of human samples alone. The specificity is ensured as a result of the original metabolic signature of OPMD pathology identified in muscle being replicated in an independent cohort. The robustness of these markers is supported by the weight of evidence of these metabolite features being consistently associated with OPMD pathology in four independent sample sets across tissues and species.

Whilst we demonstrate that our combinatorial approach has successfully identified a panel of biomarkers capable of predicting OPMD with good accuracy, our study suffers from limitations associated with the use of animal models and how representative the animal model is of the human disease. For example, in this study, the phenotype observed in the A17.1 model used is produced by overexpressing expanded PABPN1, which is different to heterozygous patients with OPMD. This raises the concerns that any biomarkers identified in the animal model may have no clinical utility; however, our results revealed a panel of nine biomarkers that were conserved between mouse and human plasma and were able to predict the presence of OPMD with good accuracy. Hence, the more accurately a model genocopies and phenocopies a human condition, the more robust the analyses generated from this workflow will be, attenuating the need for large sample cohorts of patient samples.

In conclusion, this study presents a solution for developing biomarkers in rare diseases with a comprehensive workflow. By shifting the statistical burden of biomarker discovery to pre-clinical models whilst maintaining disease specificity by only translating biomarkers from the site of pathology to the periphery, the successful identification of a panel of nine candidate biomarkers, which were validated across different species and tissues. This demonstrates the potential of the proposed workflow for biomarker discovery in rare diseases.

## Figures and Tables

**Figure 1 metabolites-13-00769-f001:**
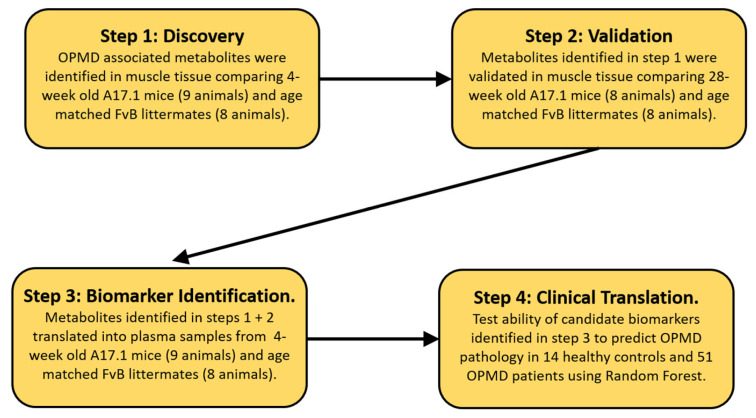
Schematic layout of the biomarker discovery pipeline utilized.

**Figure 2 metabolites-13-00769-f002:**
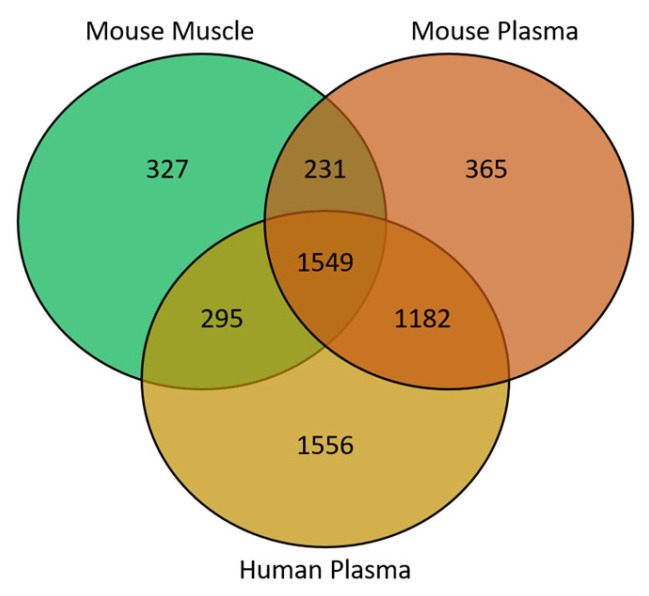
Venn diagram showing the overlap of metabolite features between sample matrices.

**Figure 3 metabolites-13-00769-f003:**
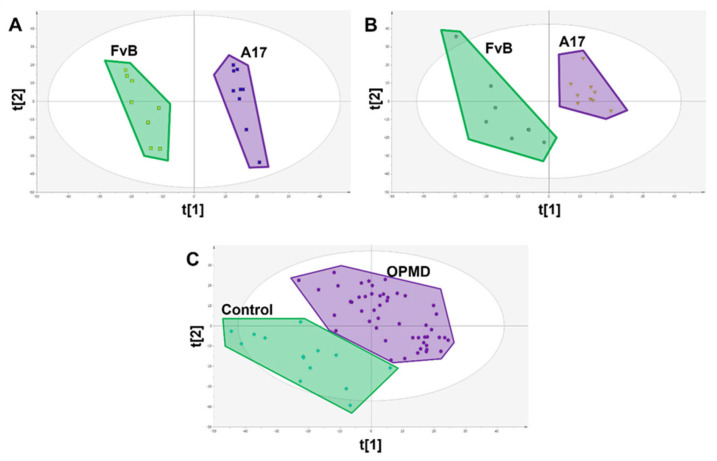
PLS-DA scores plots showing class difference based on all measured metabolite features. (**A**) Scores plot of FvB vs A17.1 mouse muscle where each spot represents a single sample (R2X = 0.385 R2Y = 0.987 Q2 = 0.928 CV-ANOVA = 4.07 × 10^−5^). (**B**) Scores plot of FvB vs A17.1 mouse plasma where each spot represents a single sample (R2X = 0.345 R2Y = 0.938 Q2 = 0.625 CV-ANOVA = 0.012). (**C**) Scores plot of FvB vs A17.1 human plasma where each spot represents a single sample (R2X = 0.386 R2Y = 0.801 Q2 = 0.651 CV-ANOVA = 2.77 × 10^−12^).

**Figure 4 metabolites-13-00769-f004:**
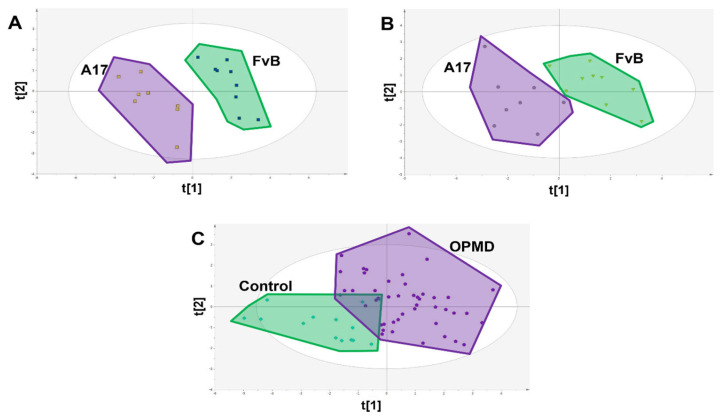
PLS-DA scores plots showing class difference based on only the 9 identified candidate biomarkers. (**A**) Scores plot of FvB vs A17.1 mouse muscle where each spot represents a single sample (R2X = 0.684 R2Y = 0.932 Q2 = 0.873 CV-ANOVA = 3.84 × 10^−5^). (**B**) Scores plot of FvB vs A17.1 mouse plasma where each spot represents a single sample (R2X = 0.615 R2Y = 0.790 Q2 = 0.662 CV-ANOVA = 0.005). (**C**) Scores plot of FvB vs. A17.1 human plasma where each spot represents a single sample (R2X = 0.544 R2Y = 0.468 Q2 = 0.345 CV-ANOVA = 5.21 × 10^−8^).

**Figure 5 metabolites-13-00769-f005:**
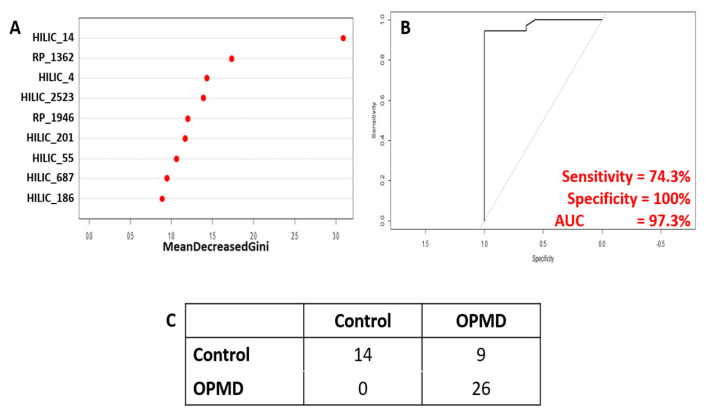
Results of random forest analysis to test classification power of identified candidate biomarkers. (**A**) Variable importance plot showing relative contribution of each CBM to achieve classification. (**B**) Receiver operating curve showing the relationship between sensitivity and specificity. (**C**) Confusion matrix showing actual vs predicted sample identities.

**Table 1 metabolites-13-00769-t001:** Clinical characteristics of human study participants.

	Control	OPMD
Sample N° (% female)	14 (50%)	51 (55%)
Age (years)	61.0 ± 12.8	60.4 ± 8.5
Mean age of onset (years)	n/a	50.5 ± 7.4

**Table 2 metabolites-13-00769-t002:** Association of candidate biomarkers with OPMD pathology in both (exploratory and replication) mouse muscle sets as well as human and mouse plasma.

	Muscle 1		Muscle 2		Mouse Plasma		Human Plasma	
	*p*-Value	FC^+^	*p*-Value	FC^+^	*p*-Value	FC^+^	*p*-Value	FC^++^
HILIC_4	0.035	0.81	0.033	0.78	0.018	0.64	0.0041	0.62
HILIC_14	8.22 × 10^−5^	0.76	0.022	0.38	0.021	0.76	1.56 × 10^−5^	0.71
HILIC_55	8.22 × 10^−5^	1.43	0.0006	2.07	0.040	0.82	0.0055	0.89
HILIC_186	0.0079	0.79	0.037	0.71	0.094	0.76	0.027	0.79
HILIC_201	0.0079	0.88	0.026	0.70	0.034	0.84	0.0073	0.84
RP_1362	0.015	0.79	0.0029	0.65	0.079	0.80	0.0023	0.72
HILIC_687	0.029	0.63	0.020	0.513	0.027	1.26	0.013	5.88
HILIC_2523	0.020	0.83	0.031	0.49	0.028	1.30	0.038	1.39
RP_1362	0.0005	0.44	0.0024	0.49	0.069	1.32	0.0073	1.74

The *p*-values were calculated using Welch’s t-test for normally distributed data and Mann–Whitney U-test for skewed data. +, fold change calculated relative to FvB; ++, fold change calculated relative to healthy controls. Metabolite features named based on the separation method used to measure them; for example HILIC_186 was measured in the HILIC analysis of the aqueous phase and RP_1362 was measured by reversed phase from the organic fraction.

## Data Availability

All processed data will be available on request from the corresponding author. Data are not publicly available due to privacy or ethical restrictions.

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
