# Peer review of "Novel Metabolomic Approach for Identifying Pathology-Specific Biomarkers in Rare Diseases: A Case Study in Oculopharyngeal Muscular Dystrophy (OPMD)"

_metabolites, 2023, doi:10.3390/metabo13060769_

Round 1

Reviewer 1 Report

The authors brought an interesting and original study with potential impact in the near future in clinical practice. Biomarkers represent challenging and potentially revolutionary features which can aid clinicians in the evaluation of clinical severity, stage of disease, follow-up, therapeutic measure (outcome), and provide a reliable and objective marker for clinicians in practice. Some points should be evaluated by the authors at this point: 

1. Gene citations must be presented in italics (e.g., PABPN1).

2. Do the authors know if any of the nine identified candidate biomarkers has been previously described in association with muscle pathologic involvement in other diseases (e.g., LGMD, miofibrillar myopathies, congenital myopathies)? Have the authors any idea about possible correlation of the biomarkers with disease severity or progression, according to their model?

Author Response

Reviewer 1

The authors brought an interesting and original study with potential impact in the near future in clinical practice. Biomarkers represent challenging and potentially revolutionary features which can aid clinicians in the evaluation of clinical severity, stage of disease, follow-up, therapeutic measure (outcome), and provide a reliable and objective marker for clinicians in practice. Some points should be evaluated by the authors at this point: 

  1. Gene citations must be presented in italics (e.g., PABPN1).

We thank the reviewer for bringing this to our attention. Where appropriate, gene nomenclature has been changed to reflect international conventions.

  1. Do the authors know if any of the nine identified candidate biomarkers has been previously described in association with muscle pathologic involvement in other diseases (e.g., LGMD, miofibrillar myopathies, congenital myopathies)? Have the authors any idea about possible correlation of the biomarkers with disease severity or progression, according to their model?

In response to the question regarding the association of the identified candidate biomarkers with muscle pathologic involvement in other diseases and their correlation with disease severity or progression in the model, we would like to emphasise that the primary focus of this paper is to present the method and workflow used for biomarker identification in rare diseases. The intention of this study was to establish the feasibility of the approach rather than delve into the specific biological functions of the identified metabolites.

While the identified biomarkers may have relevance in other muscle-related conditions such as LGMD, myofibrillar myopathies, or congenital myopathies, further research would be required to investigate these associations, including obtaining relevant patient samples. Additionally, exploring the correlation between biomarker abundance and disease severity or progression would necessitate a more comprehensive study with tailored data collection specifically designed for that purpose.

Therefore, it is important to recognise that this study serves as a “template” step in the identification of potential biomarkers in rare diseases, and future investigations should be conducted to elucidate the biological significance of these markers in different muscle-related disorders and their relationship with disease progression.

Reviewer 2 Report

The study by Harish et al. describes an innovative approach to identifying biomarkers in situations such as rare diseases, where gathering the required number of human samples is nearly impossible. There is a clear gap of knowledge that the authors are working to address, and I commend them on their handwork. In general, the paper is well-written and scientifically sound, but it needs to be improved before it can be published.

In the methods

- Please specify the number of tissue and plasma samples obtained from mice, as well as plasma samples obtained from OPMD patients, that were used in this study.

In the results

- Figure 1 should be improved both visually and in terms of the details provided. For example, there is no mention of the number of samples used in each step or the use of human plasma samples in Step 4.

- Please provide a Venn diagram to explain the metabolites that overlap the various samples. Authors may want to consider rewording the sentence from lines 241-244 to make it clearer. Again, the authors refer to table 2, which only shows 9 metabolites.

-A major concern is that the authors do not identify the candidate metabolites. How can one apply for clinics if they do not have identification? Metabolites should at the very least be putatively classified based on their structure. In addition, it would be very interesting to see if the 9-biomarker panel is linked to the molecular mechanisms underlying OPMD.

In the discussion

- This section appears to be incomplete, as the authors mention two major limitations but only one is presented. Furthermore, no study conclusion is provided.

Minor corrections:

- remove the dot at the end of the title and write “oculopharyngeal” in lower case.

- Abstracts are typically written without paragraphs.

-Line 63 it should be “to develop”

-Line 143: Please revise the subtitle to reflect the inclusion of both tissue and plasma preparation.

-Line 228: the reference to Table 2 is inappropriate here because the authors are referring to the 326 metabolites found to be significantly different between groups.

- Please note that the declaration sections at the bottom of the paper require revision.

Author Response

Reviewer 2

The study by Harish et al. describes an innovative approach to identifying biomarkers in situations such as rare diseases, where gathering the required number of human samples is nearly impossible. There is a clear gap of knowledge that the authors are working to address, and I commend them on their handwork. In general, the paper is well-written and scientifically sound, but it needs to be improved before it can be published.

In the methods

- Please specify the number of tissue and plasma samples obtained from mice, as well as plasma samples obtained from OPMD patients, that were used in this study.

We thank the reviewer for their helpful comment and have added information on animla numbers in the main text, reading as follows:

“4-week-old A17.1 OPMD model (9 animals) mice and littermate FvB healthy controls (8 animals) were fasted overnight, sacrificed, with blood collected by cardiac puncture, allowed to clot overnight at 4℃.”

Information on the number of control and OPMD patients has been added to Table 1.

In the results

- Figure 1 should be improved both visually and in terms of the details provided. For example, there is no mention of the number of samples used in each step or the use of human plasma samples in Step 4.

We thank the reviewer for their comments on the clarity of this figure and have taken their feedback on board and have amended the figure accordingly.

- Please provide a Venn diagram to explain the metabolites that overlap the various samples. Authors may want to consider rewording the sentence from lines 241-244 to make it clearer. Again, the authors refer to table 2, which only shows 9 metabolites.

We thank the reviewer for the detailed assessment of our manuscript and we have removed the erroneous reference to table 2.

We have also taken on board the reviewers comment about clarifying the overlap between sample types and have included a venn diagram to better illustrate this information, and have added additional text:

“In this study 327 of the metabolite features were unique to the murine muscle samples with 365 and 1556 features unique to mouse and human plasma respectively. In total 1549 features were common to all 3 sample types whilst 231 were shared between only mouse muscle and plasma and 1182 shared between murine and human plasma (Figure 2).”

-A major concern is that the authors do not identify the candidate metabolites. How can one apply for clinics if they do not have identification? Metabolites should at the very least be putatively classified based on their structure. In addition, it would be very interesting to see if the 9-biomarker panel is linked to the molecular mechanisms underlying OPMD.

We appreciate the reviewer's concern about the lack of identification of the candidate metabolites in our study. We apologise for not providing explicit details on the metabolites in the current manuscript. However, we would like to clarify that the focus of this paper is the development and validation of the biomarker identification workflow, rather than specifically highlighting the biological function or structural classification of the identified metabolites. We acknowledge the importance of identifying and classifying the metabolites for clinical applications. We are currently working on the structurally classifying these metabolites, and the results will be presented in a future publication dedicated to their identification and potential biological function.

In the discussion

- This section appears to be incomplete, as the authors mention two major limitations but only one is presented. Furthermore, no study conclusion is provided.

The last paragraph has been modified to add clarity and the study conclusion added.

Minor corrections:

- remove the dot at the end of the title and write “oculopharyngeal” in lower case.

- Abstracts are typically written without paragraphs.

-Line 63 it should be “to develop”

-Line 143: Please revise the subtitle to reflect the inclusion of both tissue and plasma preparation.

-Line 228: the reference to Table 2 is inappropriate here because the authors are referring to the 326 metabolites found to be significantly different between groups.

 We thank the reviewer for pointing out these issues and have addressed these issues.

Round 2

Reviewer 2 Report

The Authors have addressed all of my concerns with the original manuscript. The revised manuscript is ready for publication.